# Understanding Interviewees' Perceptions and Behaviour towards Verbally and Non-verbally Expressive Virtual Interviewing Agents

JINAL THAKKAR, International Institute of Information Technology, Bangalore, India

POOJA S. B. RAO, University of Lausanne, Switzerland

KUMAR SHUBHAM, International Institute of Information Technology, Bangalore, India

VAIBHAV JAIN, Universität des Saarlandes, Germany

DINESH BABU JAYAGOPI, International Institute of Information Technology, Bangalore, India

Recent technological advancements have boosted the usage of virtual interviewing platforms where the candidates interact with a virtual interviewing agent or an avatar that has human-like behavior instead of face-to-face interviews. As a result, it is essential to understand how candidates perceive these virtual interviewing avatars and whether adding features to boost the system's interaction makes a difference. In this work, we present the results of two studies in which a virtual interviewing avatar with verbal and non-verbal interaction capabilities was used to conduct employment interviews. We add two interactive capabilities to the avatar, namely the non-verbal gestures and the verbal follow-up questioning and compare it with a simple interviewing avatar. We analyze the differences in perception with self-rated measures and behaviour with automatically extracted audiovisual behavioural cues. The results show that the candidates speak for a longer time, feel less stressed and have a better chance to perform with verbally and non-verbally expressive virtual interviewing agents.

CCS Concepts: • **Embodied interaction**; • **Human-robot/agent interaction**;

**ACM Reference Format:**
Jinal Thakkar, Pooja S. B. Rao, Kumar Shubham, Vaibhav Jain, and Dinesh Babu Jayagopi. 2022. Understanding Interviewees' Perceptions and Behaviour towards Verbally and Non-verbally Expressive Virtual Interviewing Agents. In *INTERNATIONAL CONFERENCE ON MULTIMODAL INTERACTION (ICMI '22 Companion), November 7–11, 2022, Bengaluru, India.* ACM, New York, NY, USA, 13 pages. https://doi.org/10.1145/3536220.3558802

## 1 INTRODUCTION

Employment interviews continue to be among the most prevalent candidate selection methods [21]. Employment interviews are used to gather information about the candidate and assess the skills and characteristics to select the right candidate for the job. For example, a human resource manager may interview all job applicants to understand their skills and determine the right-fit candidate for the job opening. While this may seem like a viable option, It has a few limitations, like the human interviewer can interview only one candidate at a given time and can conduct limited interviews in a day. It is not scalable and involves expenses such as scheduling, infrastructure, and workspace, among others. Recruiters are turning to futuristic alternatives like social recruiting and video interviews to save expenses and reduce hurdles [44]. Hirevue [13] and Recright [32] are among the few companies that have commercialised these virtual interviewing platforms.

Asynchronous video interviews (AVI) have become popular for preliminary screening and interview coaching. Automatic interview and coaching systems mimic the behaviour of an interviewer assisting in simulated interviews. When compared to in-person interviews, the practicality and convenience of automatic AVI evaluation is promoting the system's widespread implementation [30]. The addition of intelligent virtual agents to AVIs makes the experience more engaging and immersive [43]. They provide a social component to the mechanical video interviewing platforms. Attempts have been made to enhance these agents' capabilities to make them more interactive. These approaches, among others, include the incorporation of non-verbal behaviour (NVB) and verbal behaviour (VB). They are significant components of believable behaviour [5]. These behaviours have been sought to be introduced to the agents almost since their inception [7][16]. With recent advances in technology, these behaviours in the agents have evolved. From incorporating social competencies and richer multimodal non-verbal behaviours [6, 49] to dynamic verbal follow-up questioning and probing [42] [28], these behaviours intend to make the agents more interactive and conversational.

With the growing momentum of AVIs and usage of virtual interviewing agents with VB and NVB behaviours, it raises an important research question: *Does addition of verbal and non-verbal behaviours to the virtual interviewing agent have an impact on interviewees?* To answer the above question, we conducted comparative studies with 30 participants taking both the interviews. As we were interested in understanding the individual effect of the verbal and non-verbal capabilities of the interviewing agent on the candidate's behaviour and perception, we conducted two comparative studies. i) **control v/s NVB**: to understand the effect of non-verbal capabilities ii) **control v/s VB**: to understand the effect of verbal capabilities. More specifically, the main contributions of this paper are: 1) We create a dual setup of a virtual interviewing platform with a virtual human avatar consisting of verbal and non-verbal capabilities. 2) We conduct separate studies of the candidates taking interviews when subjected to an interviewing agent with a) ability to perform certain non-verbal gestures b) ability to generate dynamic follow-up questions in comparison to an agent with no additional abilities, and finally 3) We analyse the differences in perception (via self-reported measures) and behaviour (via automatically extracted features) of the candidates in both the settings. To the best of our knowledge, there has not been a study that compares the candidate experiences in the virtual interviewing platforms consisting of the interviewing agent with and without the verbal and non-verbal behaviours.

## 2 RELATED WORK

There have been previous attempts to use virtual agents with different attributes in different scenarios. For example, in an interviewing scenario, the experimental study of an automated conversational coach - MACH [23] has shown that the use of non-verbal gestures in virtual agents can be used effectively. TARDIS [1] has built a scenario-based serious game simulation platform to support social training and coaching in the context of job interviews for young people who are unemployed, uneducated or untrained. Intelligent Multimodal virtual agents named PARLEY [35] is also be used to train users in difficult social situations. ISI, a visual interaction agent which helps promote verbal communication skills in children. Previous studies have shown that the candidates are not at disadvantage when they appear for virtual agent based interviews in comparison to the face-to-face interviews [24, 37]. Rasipuram et al. [30] also supports the use of virtual interviewing agents as equally good as the face-to-face interviews when assessing the communication skills of the candidates for employment interviews. Wang and Ruiz [50] have highlighted the importance of non-verbal behavior in virtual agents to emulate expressivity and multimodality. In their literature review, they conclude that though virtual agents with NVB have been successful in improving users' perceptions, there have also been some inconclusive results. Sproull et. al [38] found that the participants attributed more personality traits when they interacted with an agent with a speaking human face than a computer system with displayed text. Virtual agents such as Rhea, a

virtual real estate agent [8] and Greta, a multi-functional virtual agent assisting applications ranging from interviews to coaching [26] have highlighted the use of non-verbal gestures in virtual agents along with speech. While the natural language integration with the virtual agent dates back to several decades [36] and have found applications like product recommendations [3]and dialogue systems [55], giving verbal conversational abilities to the virtual agent is evolving with the major trends in natural human–computer interfaces. Ran Zhao et. al. [56] have identified building rapport as an important part of building human interactions, virtual agents with verbally expressive behaviour will help in building a rapport with the user. Karolina Kuligowska [17] reported that the biggest challenge in designing a good chatbot was to develop a mechanism for a contextual dialogue flow. Most the commercially available Polish-speaking chatbots were rule-based and lacked natural language processing. The chatbots that could lead a coherent dialogue, handle complex user inputs were rated better. Although, there have been studies and attempts to make the virtual agent as human-like as possible for specific applications, to the best of our knowledge, there has not been a user study that addresses how the interviewees perceive the virtual agents with verbal and non-verbal abilities. Our works attempts to close these research gaps.

## 3 TOOLS DESIGN

We developed a custom tool with a virtual interviewing agent to conduct the two comparative studies. For both the settings, we used the ICT Virtual Human Toolkit (VHToolkit) [48] to build the interviewing agent. The VHToolkit is used as an embodied conversational agent which we have customised to act as the interviewing agent since it gave nearly full control over the virtual avatar. The VHToolkit is a collection of modules, tools and libraries which helps create interviewing agent. There are 5 major process and modules which help in creating the conversational interviewing agent namely User Multimodal Analysis (Multisense) [41] , Dialogue Manager (NPCEditor) [20], Behavior Planning and Sequencing (NVBG) [19], Behavior Realization (SmartBody) [45], Rendering (vhtoolkitUnity) [47]. Please refer to the figures in appendix A for sample illustrations of the avatar.

### 3.1 Control setup

For the controlled setup, there are no commands sent for the virtual avatar to show any gestures. The interview consists of six hard-coded questions being asked to the candidates. This setup is used in the first phase of the interview in both the studies (refer section 4.1.2). Every question is customised into a VHMsg [46] .

### 3.2 Virtual Agent with Non-Verbal Gestures (NVB)

Since the open source version of vhtoolkit module does not generate any beat or metaphoric gestures during conversation, We developed our own behavior generation module. We have generated three types of gestures: Metaphoric, Deictic, and Beat Gestures to give non-verbal gesture generation capabilities to the interviewing agent. We model the non-verbal behaviours of our avatar only through physical gestures, and not through phonetic expression. It is because the voice modulation from the TTS service available in VHToolKit were not satisfactory. We use a suite of pre-animated gestures available in VHToolkit to display the non-verbal gestures. The animation selection and synchronization process is based on the architecture presented by Ravenet, Brian, et al. [31]. Cienki and Müller [10] concluded that the Image Schemas can be used to characterize gestures hence we used them to communicate verbal to non verbal channels to generate metaphoric gestures. We parse the surface text of the question to be asked to generate a SYNSET [25] (SurfaceTextSynset) of each word using the WordNet dictionary [54], and disambiguate the meaning of each word using the LESK method [52]. This is used to generate Hypernyms [25]. We compare the similarity of SurfaceTextSynset and

its Hypernyms with the list of the Synset of our list of Image Schemas, and assign an Image Schema to the words if similarity is found which is used for animation mapping and a custom VRSpeak message [14] is generated and sent to the VHToolKit. Figure [2] shows the Metaphoric Gesture Generation pipeline. To generate Beat Gesture, we referred to a study by L. Wang et. al. [51] which concluded that the critical words in a spoken sentence are accompanied by a beat gesture. We used Rapid Automatic Keyword Extraction Algorithm (RAKE) [33] to extract "key" words from the surface text and assigns an "importance" score to the extracted keywords/phrases. We assign a BEAT gesture to a word in our surface text if its importance-score crosses the threshold value of 1.0 which we found after experimentation with different values and scenarios. The Deictic Gesture Generator draws its similarity with the 2006's NVBG for ECA [19]. For each gesture, a communicative function is defined which is mapped to a set of certain words. When one of these words appear the communicative intent is triggered to generate a Deictic Gesture.

### 3.3 Virtual Agent with Follow-up Question Generation (VB)

The follow-up questions generated and integrated into the VHToolkit were adapted from the module developed by Rao S. B. et al. [28]. They define a follow-up question as the one that is dynamically generated depending on original interview question and the user input in the form of answer. The follow-up question generation model uses a Generative Pre-trained Transformer (GPT-2) [27], fine-tuned on the asynchronous interview dataset, released publicly in the same work. The dataset has over 1000 triplets of question, answer and follow-up. These triplets are embedded and concatenated in order to form an input for the model during training. We use the same procedure as described in the paper to train the follow-up question generation model[1]. The followup question hence generated is converted into a Behavioral Markup Language (BML) under the speech element. As stated in [28], we restricted the follow-up question to one level of limited probing. The avatar then asks the followup question to the user. Of the six questions posed to candidate, every alternate question is a followup question.

## 4 METHODOLOGY

### 4.1 Experimental Setup

The experiment was conducted as two separate within-subject experiments, control vs VB and control vs NVB. The same group of participants took the interviews in both the control and the experimental setting in each study. The experiment was conducted in four phases the preparatory phase, first phase, second phase and the concluding phase. All the interviews were conducted over a Zoom call [57] which was recorded to be analysed later with former consent from the candidate. The inbuilt camera and microphone in the candidate's laptop or phone was used to capture the video and audio of the candidate. The candidates selected for the interview were English-speaking graduate students or working professionals. The average age of the candidates is 25.7 years and the standard deviation is 3.1 years. The candidates had some experience either in terms of working at a company or an internship or experience of working in a team. There were 8 females and 22 males in both the VB and NVB groups. The candidate always start with the preparatory phase, although the order of the first phase and second phase was completely randomized.Of the 30 participants, 15 participants took the NVB/VB interview first followed by a controlled interview and vice versa for the rest of the 15 candidates. The candidate then ends the process with the concluding phase.

*4.1.1 Preparatory Phase.* Before appearing for the interviews, the candidates signed a consent form to permit the use of their data. The candidates were briefed on how to use the interface. They were instructed to assume as if they were

---

[1]https://github.com/poorao/followQG

appearing for a real job interview. Hence a make believe job description scenario was presented to them. The candidates then appeared for the interviews in the first and second phase in random order.

*4.1.2    First Phase.* The interview in this phase is a controlled interview where the set of questions are hard-coded for every candidate. This phase is common in both control v/s NVB and control v/s VB. The six questions asked during the interview fall broadly into self-introduction (Q1: The candidate is asked to introduce themselves), past behavior questions (Q2, Q3, Q4: The candidate is asked questions related to their past experiences where they were a part of a disagreement or failed at a task and how they managed to handle the situation.) and finally the category of future aspirations (Q5, Q6: The candidate is asked questions related to their future career goals). The ordering of the questions except the first question i.e. the self-presentation question, was randomized. At the end of the interview, the candidate was asked to fill the post-interview questionnaire. These questions were selected to probe the past, current and the future scope details of the candidates, thus giving them a chance to explain themselves elaborately. (more details in section 4.2.1.)

*4.1.3    Second Phase.* The second interview for the candidate could either be the virtual interviewing agent with verbal capability in control v/s VB study or non-verbal capability in control v/s NVB study. In control v/s VB study, every alternate question was a follow-up question (Q2, Q4, Q6). The remaining three questions fell in the categories of self introduction (Q1), past behavior questions (Q3) and future aspirations (Q5). Every alternate question was chosen to be a follow-up question as restricted probing would assist in finding the right balance between structure of the interview and conversational interaction [15, 28]. The agent did not produce any non-verbal gestures. In control v/s NVB study, all the six questions were hard-coded and were of the same categories and a similar difficulty level as the first phase. The agent was capable of producing non-verbal gestures based on the question in control v/s NVB. Candidates in both studies fill a post interview questionnaire after the interview.

*4.1.4    Concluding Phase.* A final feedback form was presented to the participants asking for their preferred interviewing method from the first and the second phases or both or none. It also consisted of open ended questions asking for the reason for their preferred interview and if they noticed any differences between the two phases.

## 4.2    Measures

*4.2.1    Self-reported Measures.* The candidate fills up a post interview questionnaire based on their interview experience. Candidates rated their experience on a scale of 1 (strongly disagree / worst ) to 5 (strongly agree / best) on different questionnaires. The post interview questionnaire had questions related to chance to perform [18], if they felt stressed, anxious , engaged and confident [22]. The chance to perform metrics helps the candidate evaluate whether the interview gave enough opportunity to show their skills and abilities or if they were able to really demonstrate if they have the required skills for the job etc. There were six questions asking if the interview gave enough chance to perform based on Bauer et al. [2]. There were questions to measure the amount of communication anxiety, behavioural anxiety and performance anxiety felt during the interview. These metrics were measured by asking questions such as, whether they got so anxious that they had trouble answering the questions, whether they felt their verbal communication skills weren't strong enough, whether they felt sick in their stomach. There were about 17 questions measuring anxiety during the interview. The final feedback form asked the candidate their choice of interview and also asked if they liked both or none of the interviewing methods.

*4.2.2 Behavioural Measures.* Multiple audio features were automatically extracted from recorded videos to account for behavioural differences within both the interviews. As a pre-processing step before carrying out the analysis of the interviews, we extracted segments only where the interviewee is answering the questions. The prosodic features such as loudness, spoken time, pitch [30] reflects multiple social traits (e.g. stress, engagement and other behavioral traits). The prosodic features help us understand the features like audio style, tone, degree of stress of the candidate. We used features like pitch, loudness and energy as a part of prosodic features extracted using OpenSmile [12]. These features have association to stress as per the recent studies [9]. Speech features like the total time of the interview, speaking rate (number of syllable/ duration), articulation rate (number of syllable/ phonotation time) were extracted. In our experiment, we used PRAAT [4] to extract these speech related features.

## 5 ANALYSIS

We have considered both the self-reported measures from the questionnaire and the behavioral features from the interviews for analysis. We have used the Shapiro Wilk test [53] for testing the normality of the features. The Wilcoxon Signed-Rank Test [39] was used for non-Gaussian distributions. For the Wilcoxon Signed-Rank test, we have included zero-differences in the ranking process and split the zero rank between positive and negative ones. We have calculated the one tailed Wilcoxon Signed rank test values in-order to get the direction for the results. The positive value for the W-Value indicates that the values of the feature obtained for the VB or NVB interviews are greater than the values for the controlled interview. The negative value indicates that the values of the feature obtained for VB/NVB interview are lesser than the control. The one tailed Paired T-Test [40] was used for Gaussian distributions of the features. We have reported the results for only significant p-values.

### 5.1 Results - control v/s NVB

*5.1.1 Results for Behavioral measures.* The total time and spoken time of the candidate was statistically higher in the NVB setting compared to the interview without the non-verbal gestures. The candidates expressed themselves more when the virtual interviewing agent had non verbal gestures. The candidates reported that *"Avatar felt more lively"*, *"There was a little bit more natural behavior"*. There is not much of a difference in the speaking rate and the articulation rate of the candidates during both the interviews. Interestingly, the mean energy of the candidate in the controlled interview is more than the energy in the NVB interviewing setting. We could not find any significant difference in the other prosodic features like pitch and loudness.

*5.1.2 Results for Self-Reported measures.* Candidates felt that they had better chance to perform in the NVB setting compared to the controlled interview. Both the stress and engaged measures showed statistically significant difference between the NVB and control settings. The more human-like gestures in the virtual avatar might have made the candidates feel less stressed, keeping them engaged during the interview. The candidates reported that they found the avatar with NVB features *"Engaging and friendly"*, *"The Interview was much more comfortable"*, *"It was quite more engaging. I was able to express more about my work, potentials, goals. I was able to connect things."* The candidates also felt less performance anxiety and less behavioral anxiety in the interview with the virtual interviewing agent with the non-verbal gestures. Also, the candidates felt equally confident in both the interviewing methods. The candidates felt they could communicate equally well in both the settings. From the final feedback form, 16 candidates preferred the interview with the virtual interviewing agent having a non-verbal gesturing capabilities and five candidates preferred

Table 1. Results of statistical tests for control v/s NVB and control v/s VB

| Feature | Control v/s NVB | | | Control v/s VB | | |
|---|---|---|---|---|---|---|
| | Test | W/T-Value | P-Value | Test | W/T-Value | P-Value |
| Total time | W | 112.0 | 0.0113* | W | 46.5 | 0.00010*** |
| Speaking Rate | T | | | T | 1.565 | 0.06437+ |
| Articulation Rate | T | | | W | | |
| Spoken Time | W | 97.0 | 0.0046** | W | 74.0 | 0.00095*** |
| Mean Pitch | W | | | T | | |
| Mean Loudness | W | | | T | | |
| Mean Energy | W | -296.0 | 0.0448* | W | | |
| Chance to perform | W | 145.0 | 0.0355* | T | 1.386 | 0.0881+ |
| Stress | W | -321.0 | 0.0318* | W | -332.5 | 0.0181* |
| Engaged | W | 115.5 | 0.0071** | W | | |
| Confident | W | | | W | 142.0 | 0.0290* |
| Communication Anxiety | W | | | T | | |
| Performance Anxiety | W | -315.0 | 0.0434* | W | | |
| Behavioral Anxiety | W | -317.5 | 0.0384* | W | | |
| Overall Anxiety | T | -2.179 | 0.0297* | T | | |

$p = 0.10^+, p \leq 0.05^*, p \leq .01^{**}, p \leq 0.001^{***}$; W - wilcoxon signed-rank test, T - paired t-test

the control. Four candidates did not have any preference and five candidate preferred both the interviewing methods equally. More details on this can be found in appendix D table 6.

## 5.2 Results - control v/s VB

*5.2.1 Results for Behavioral measures.* The total time and speaking time of the candidate in the VB setting is statistically different from the control setting. Since the followup question probed more information about the previous question, candidates perhaps may have had a longer conversation and provided more content serving the purpose of a follow-up as per its definition [15, 28]. One of the candidates reported that he was able to express more in the interview with VB capabilities. The speaking rate of the candidates in the VB setting was more compared to the control. This suggests that the candidates spoke faster in the interview with the follow-up question generation capabilities, possibly informing that they were more involved in this interview setting as high involvement conversational styles is characterized by fast speech rate [11]. There wasn't a significant difference in the prosodic features like the pitch, loudness and energy.

*5.2.2 Results for Self-Reported measures.* The candidates reported that they had better chance to perform in the VB setting. The candidates felt more confident in the interview with the virtual agent asking the followup question. The candidates felt relatively less stressed in the interview with the followup question. The anxiety levels were statistically not different in both the interviewing methods. The candidates felt equally engaged in both the interviewing methods. As per the final feedback, 16 candidates preferred the interview with the virtual interviewing agent having a follow-up question generation capabilities v/s six candidates preferring the controlled interviewing method. Four candidates did not have any preference and four candidates preferred both the interviewing methods equally.

## 5.3 Correlation Analysis

In this subsection, we present the results of correlation analysis between automatically extracted behavioural features and self-reported measures to understand the relationship between them. We only report correlations that are significant. For details on all the correlation values, please refer to the tables in appendix B. For the NVB group, we found that

the confidence of the candidate was positively correlated to the mean pitch and mean loudness. This overall indicates that the confident candidates spoke clearly. The performance anxiety and overall anxiety was positively correlated with the articulation rate. These results are in line with the prior literature [29, 30] where such prosodic features are used to relate with hirability measures. For the VB group, we found that the engagement and confidence was slightly negatively correlated with total time and spoken time. Behavioural anxiety was positively correlated with the speaking rate and spoken time of the candidate. This is slightly in contrast with the results in section 5.2. However, probing and follow-up inquiries can make interviews more challenging [15], resulting in a minor decline in confidence and an increase in anxiety when candidates speak more to clarify responses.

## 5.4 Qualitative Analysis

We performed a manual qualitative analysis of the open-ended user responses from the final form. We did initial coding to extract the important topics [34]. We provided anecdotal evidence that could support our quantitative results in the above sections and in the following. Candidates in NVB setting preferred the controlled interview because they found it comfortable, the questions felt better and less ambiguous. Candidates who preferred the NVB interview stated that the interviewing method was engaging, comfortable, lively and interactive. One of the candidates said "*Setting 1 felt like literally talking to a bot. In setting 2, avatar felt more lively.*" Although they felt that the questions were in-depth and a little difficult than the control. Candidates in VB setting who preferred the controlled interview stated that the interview was more friendly and comfortable since there was no further questioning or feedback from the interviewer. Candidates who preferred the VB interview stated that they felt the interview was more about the candidate itself, their goals, accomplishments, opportunity to show their skills, strengths and weaknesses and speak more about themselves. The questions felt more relevant, had a flow and were interesting compared to the control. To quote one of the candidates, *"First method was just a few standard set of questions, anybody can come with a standard set of answers and do just fine in the interview, whereas in the second one, it was interactive and I could express who I really am".* Although, one of the candidates felt under-confident because of some unexpected questions. Some candidates expected more structure to the interview and quoted *"The questions felt a little personal and instead they should be more professional and should have a structure to the questions."*

## 6 CONCLUSION

In this paper, we have systematically studied the effects of adding verbal or non-verbal behaviour on the virtual interviewing agent on the interviewees'. We conclude from the results that the candidates feel that they performed better when the virtual avatar has these features. The candidates spoke more and are able to express themselves better in the interviews with the avatar emulating human-like behaviour. We observed that the candidates are relatively less stressed with the virtual interviewing avatar with verbal or non verbal cues. Non-verbal gestures of the avatar helped reducing the anxiety levels of the candidates while appearing for the interviews. Of course, these non-verbal gestures are still basic in manifestation, more research in improving these may help in improving the candidate experience. Candidates felt more confident with the verbal behaviour in the avatar compared to control, but they also felt slightly less confident as they spoke more and were questioned. The avatar does not display any listening behaviour while the candidate is answering. Adding this is feature may make the avatar even more human like. The follow-up question considers only the previous answer, taking all the previous answers and the context into consideration will help in probing relevant information from the candidate. The current limitation of this study is that it does not compare the effect verbal and non-verbal features together on the candidates. We intend to do this study in the future.

## 7  ACKNOWLEDGMENT

Jinal is funded by International Institute of Information Technology, Bangalore, Institute Fellowship and Vaibhav Jain and Kumar Shubham is funded by Openstream.ai. We would like to thank all the participants who contributed for data collection.

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

## A  VIRTUAL AVATAR

Figure 1 illustrating virtual interviewing agent with and without non-verbal gestures.

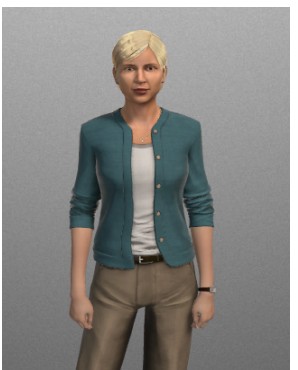 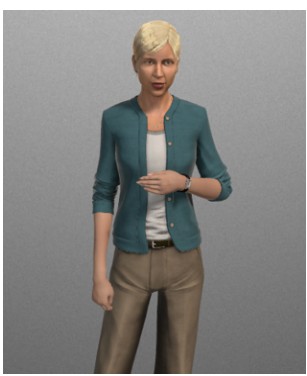

Fig. 1. A Simple Virtual interviewing Agent and A virtual Interviewing Agent illustrating Non-Verbal Gesture

## B  SPEARMAN'S CORRELATION

The below tables show the spearman's correlation coefficients for all the interview settings.

## C  METAPHORIC GESTURE GENERATOR PIPELINE

The below figure shows the metaphoric gesture generator pipeline.

| Type | Feature | total_time | speaking_rate | artn_rate | spoken_time | mean_pitch | mean_loud | mean_energy |
|---|---|---|---|---|---|---|---|---|
| FOLLOWUP | chance_to_perf | | -0.203 | -0.337 | -0.209 | -0.361 | -0.309 | |
| FOLLOWUP | stress | | | | | | -0.327 | |
| FOLLOWUP | engaged | -0.206 | | | -0.318 | | | |
| FOLLOWUP | confident | -0.314 | | | $-0.294^*$ | | | |
| FOLLOWUP | comm_anxiety | | | 0.215 | | | | |
| FOLLOWUP | perf_anxiety | | | | | | | |
| FOLLOWUP | behave_anxiety | | 0.321 | | 0.294 | | | |
| FOLLOWUP | anxiety | | | | $0.242^+$ | | | |

$p = 0.10^+, p \leq 0.05^*, p \leq .01^{**}, p \leq 0.001^{***}$

Table 2. Spearman's correlation between self-reported and behavioural features for(Controlled) Verbal Setting

| Type | Feature | total_time | speaking_rate | artn_rate | spoken_time | mean_pitch | mean_loudness | mean_energy |
|---|---|---|---|---|---|---|---|---|
| Followup | chance_to_perf | | -0.2027 | $-0.33713^+$ | -0.20945 | $-0.36054^+$ | -0.30861 | |
| Followup | stress | | | | | | $-0.32686^+$ | |
| Followup | engaged | -0.20628 | | | $-0.31816^+$ | | | |
| Followup | confident | $-0.31423^+$ | | | -0.29422 | | | |
| Followup | comm_anxiety | 0.270372 | | 0.215476 | | | | |
| Followup | perf_anxiety | | | | | | | |
| Followup | behave_anxiety | | $0.32069^+$ | | 0.29403 | | | |
| Followup | anxiety | | | | 0.241991 | | | |

$p = 0.10^+, p \leq 0.05^*, p \leq .01^{**}, p \leq 0.001^{***}$

Table 3. Spearman's correlation between self-reported and behavioural features for Verbal Setting

| Type | Feature | total_time | speak_rate | artn_rate | spoken_time | mean_pitch | mean_loud | mean_energy |
|---|---|---|---|---|---|---|---|---|
| Controlled - NVBG | chance_to_pf | | | | 0.21 | 0.252 | | |
| Controlled - NVBG | stress | 0.205 | | | 0.217 | | | |
| Controlled - NVBG | engaged | | -0.211 | | | -0.282 | -0.25 | |
| Controlled - NVBG | confident | | 0.22 | -0.221 | $0.389^*$ | | | $0.372^+$ |
| Controlled - NVBG | comm_a | | | | | -0.222 | | -0.245 |
| Controlled - NVBG | perf_a | | | $0.365^+$ | $-0.336^+$ | | | $-0.41^*$ |
| Controlled - NVBG | behave_a | | | | | -0.256 | -0.243 | |
| Controlled - NVBG | anxiety | | | $0.206^+$ | -0.34 | -0.207 | | $-0.339^+$ |

$p = 0.10^+, p \leq 0.05^*, p \leq .01^{**}, p \leq 0.001^{***}$

Table 4. Spearman's correlation between self-reported and behavioural features for Controlled (Non-Verbal) Setting

| Type | Feature | total_time | speaking_rate | artn_rate | spoken_time | mean_pitch | mean_loud | mean_energy |
|---|---|---|---|---|---|---|---|---|
| NVBG | chance_to_perf | $-0.372^*$ | | -0.224 | -0.264 | -0.214 | | |
| NVBG | stress | | | 0.251 | | -0.249 | -0.255 | |
| NVBG | engaged | | | -0.211 | | | | -0.209 |
| NVBG | confident | | 0.304 | $-0.418^*$ | 0.261 | 0.301 | $0.417^*$ | 0.295 |
| NVBG | comm_anxiety | | | 0.316 | | $-0.336^+$ | $-0.321^+$ | |
| NVBG | perf_anxiety | | | $0.464^*$ | | | -0.273 | |
| NVBG | behave_anxiety | 0.269 | | 0.271 | 0.22 | | -0.212 | |
| NVBG | anxiety | | | $0.461^*$ | | -0.21 | $-0.358^+$ | |

$p = 0.10^+, p \leq 0.05^*, p \leq .01^{**}, p \leq 0.001^{***}$

Table 5. Spearman's correlation between self-reported and behavioural features for Non-Verbal Setting

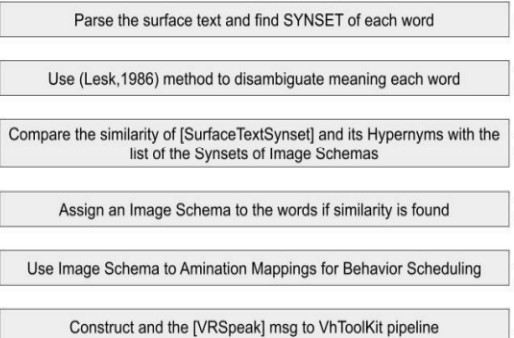

Fig. 2. Metaphoric Gesture Generator pipeline

## D  INTERVIEW TYPE PREFERENCES

| | | |
|---|---|---|
| VB v/s Control | VB | 16 |
| | Controlled | 6 |
| | Both | 4 |
| | None | 4 |
| NVB v/s Control | NVB | 16 |
| | Controlled | 5 |
| | Both | 5 |
| | None | 4 |

Table 6. Interview Preference of Candidates

