# OpenReview forum: "Understanding Interviewees’ Perceptions and Behaviour towards Verbally and Non-verbally Expressive Virtual Interviewing Agents"
_ACM.org/ICMI/2022/Workshop/GENEA — GENEA Challenge & Workshop 2022 Workshopproceeding_

### Official Review · Reviewer_rtb3 · 2022-08-16

**Rating:** 7
**Confidence:** 3

**Review:**

This paper presents results of a study conducted on interviewees behavior toward an interviewing virtual agent that uses both non-verbal and verbal behavior cues.  The authors set up a platform with an avatar with verbal and non-verbal capabilities, conduct experiments where subjects conduct interviews with agents capable of non-verbal behaviors and capable of generating follow-up questions, and analyze users' self-assessed perceptions of the interaction.

The paper cites a lot of relevant related work, demonstrating how this work is situated relative to the rest of the field.  One other work the authors should compare to (although the domain is not interviewing) is:

• Wang, H., Gaddy, V., Beveridge, J. R., & Ortega, F. R. (2021). Building an emotionally responsive avatar with dynamic facial expressions in human—computer interactions. Multimodal Technologies and Interaction, 5(3), 13.

The methods are described quite clearly and executed with appropriate control conditions.  There some issues with self-reporting, of course, and I wonder if the authors can related thee results reported using the self-reporting metrics with the more direct quantitative results (e.g., from the top half of table 1), to validated what emerged from the self-reported data.  However, overall, the paper is well-written and presented with both quantitative and qualitative analysis with examples, which I appreciate.

One small issue: The title doesn't seem grammatical to me: "Understanding Interviewees’ Perceptions and Behaviour to Verbally and Non-verbally Expressive Virtual Interviewing Agents" - it seems like "to Verbally" should be "toward Verbally"

One big issue: This has less to do with the technical content of the paper and more with the premise of the entire paper, and pertains to larger issues in the field as a whole -- the situation here is a virtual interview, where the avatar at least notionally is not a peer to the human subject, but holds a position of power.  Should virtual interviewers even be a thing we as a community should encourage though research?  Does presenting a potential hire with a fake human instead of a real human interviewer not dehumanize the interviewee from the get go?  This strikes me as a way to reproduce old, exploitative patterns of the labor market at larger scale, and seems like a bigger issue than whether the avatars gestures or not.  Do the authors consider some of these ethical issues with the motivation beyond the technical capabilities?

---

### Official Review · Reviewer_CdjV · 2022-08-18
**Solid perception study re verbal and non-verbal expressiveness in virtual humans**

**Rating:** 7
**Confidence:** 5

**Review:**

The paper presents two experiments, where users participate in zoom-interviews with a virtual human, with or without non-verbal expressions or extra verbal embellishments (follow-up questions), and study how these affect behavioural measures as well as self-reported measures of the users. Results show significant effect of both types of modifications.

The paper is very well-written and easy to follow. The experiments are presented in a clear and structured way, and the results are clearly presented.

Regarding the tool pipeline: Beat gestures are generally co-occurring with prominent/stressed words in the speech. it is stated that generation of beat gestures is based on RAKE score extracted from the text - how does this compare to the actually realized TTS output in VHToolkit? (It would seem naiive to assume that the TTS system and RAKE always would choose to emphasize the same words), please add some clarification or comment re. this.

A couple of possible manuscript improvements:

4.1 I would be good to mention at the onset of this section that the experiment really consists of two separate within-subject experiments, control vs VB and control vs NVB.
4.2.2 some missing words in first sentence

---

### Decision · Program_Chairs · 2022-08-20

**Decision:**

Accept (Workshop proceeding)

**Comment:**

Reviewers agree that the paper describes the systems well, and has only minor issues. Most notably, authors should address the ethical concerns around this work and clarify how beat gestures were placed. See below for the full reviews.

We suggest that the authors carefully consider the feedback received from the reviewers and use it to improve their manuscript for the camera-ready submission.